# Fully stretchable active-matrix organic light-emitting electrochemical cell array

Jia Liu[1,6], Jiechen Wang [2,6], Zhitao Zhang[1,6], Francisco Molina-Lopez [1], Ging-Ji Nathan Wang[1], Bob C. Schroeder [1], Xuzhou Yan [1], Yitian Zeng[2], Oliver Zhao[2], Helen Tran[1], Ting Lei [1], Yang Lu[3], Yi-Xuan Wang[1,4], Jeffrey B.-H. Tok [1], Reinhold Dauskardt[2], Jong Won Chung [5], Youngjun Yun [5✉] & Zhenan Bao [1✉]

Intrinsically and fully stretchable active-matrix-driven displays are an important element to skin electronics that can be applied to many emerging fields, such as wearable electronics, consumer electronics and biomedical devices. Here, we show for the first time a fully stretchable active-matrix-driven organic light-emitting electrochemical cell array. Briefly, it is comprised of a stretchable light-emitting electrochemical cell array driven by a solution-processed, vertically integrated stretchable organic thin-film transistor active-matrix, which is enabled by the development of chemically-orthogonal and intrinsically stretchable dielectric materials. Our resulting active-matrix-driven organic light-emitting electrochemical cell array can be readily bent, twisted and stretched without affecting its device performance. When mounted on skin, the array can tolerate to repeated cycles at 30% strain. This work demonstrates the feasibility of skin-applicable displays and lays the foundation for further materials development.

[1] Department of Chemical Engineering, Stanford University, Stanford, CA 94305, USA. [2] Department of Materials Science and Engineering, Stanford University, Stanford, CA 94305, USA. [3] Beijing National Laboratory for Molecular Sciences (BNLMS), College of Chemistry and Molecular Engineering, Peking University, 100871 Beijing, China. [4] Tianjin Key Laboratory of Molecular Optoelectronic Sciences, Department of Chemistry, School of Science, Tianjin University, 300072 Tianjin, China. [5] Samsung Advanced Institute of Technology, Samsung Electronics, Yeongtong-gu, Suwon-si, Gyeonggi-do 16678, South Korea. [6]These authors contributed equally: Jia Liu, Jiechen Wang, Zhitao Zhang. ✉email: youngjun.yun@gmail.com; zbao@stanford.edu

Tissue-like electronics, enabled by soft electronic materials and biomimetic structures, were shown to be important for consumer electronics[1,2], wearable electronics[3–5] and biomedical devices[6–10] due to the possibility of their seamless integration with the human body and biological systems. Recently, using intrinsically stretchable materials to mimic and extend the mechanical properties and functionalities of skin, a variety of fully stretchable electronic components have been demonstrated, such as pressure sensors[11,12], temperature sensors[13] and analogy circuits[14,15]. Despite these advancements, fully stretchable skin displays have not yet been reported. If realized, it will offer opportunities for direct visual interaction and feedback for human-electronics interface.

To enable a fully stretchable skin-like display, two important components are needed: intrinsically stretchable light-emitting device array and intrinsically stretchable thin-film transistor (TFT)-driven active matrix. In addition, stretchable interconnects capable of connecting individual light-emitting pixel with the switching transistor vertically, while maintaining the overall stretchability of the monolithic device is critical to gain enhanced spatial resolution and contrast, as compared to passive-matrix devices[16]. To date, the reported stretchable displays that are either a passive-matrix-driven stretchable light-emitting device[17–21] or a stretchable transistor manually connected to a rigid light-emitting diode (LED)[22,23]. Some reports[24] leveraged the small strain tolerance up to 5% of flexible but non-stretchable electronic and optical materials (e.g. polycrystalline silicon and commercially available OLED) to fabricate AMOLED. Their stretchability is, however, impossible to be further improved due to the limited stretchability of the flexible electronic and optical materials.

Here, we present robust materials and processes that afford a fully stretchable skin-like display. Specifically, this was accomplished by integrating fully stretchable TFT array and stretchable organic light-emitting electrochemical cells (OLECs) to yield a fully stretchable transistor-driven active-matrix OLEC array (AMOLEC) array. Notably, even though the commercial AMOLED display active matrix consists of two transistors and a capacitor, previous demonstrations of new types of TFTs for driving displays have typically started with one switching TFT[16,25,26].

## Results

**Stretchable, chemically orthogonal photopatternable dielectrics.** One of the challenges in this work is the development of high-performance stretchable transistor arrays that provide the required current density to operate the stretchable OLEC. The typical current density needed to operate a stretchable LEC device is around 2–2.5 mA/cm². To drive a typical stretchable OLEC pixel for visualization (e.g. 2 × 2 mm for one pixel), the individual stretchable TFT needs to provide a relatively large current (~10$^{-4}$ A) at moderate voltages[17]. While there are several reported stretchable OTFTs, their drain currents were observed to be mostly <10$^{-6}$ A even at high gate voltages[14,15]. This is primarily due to the channel length, $L$, of the source/drain (S/D) electrodes, which are largely limited to ca. 100-μm resolution from the spray-coating of the stretchable conductors as well as relatively thick stretchable gate dielectric layer. To address this issue, one solution is to substantially reduce the thickness of stretchable gate dielectric layer and increase the active area of the intrinsically stretchable transistor, which increases the ratio of channel width vs. channel length ($W/L$) to enhance the drain current. In a typical transistor design that includes the interdigitated S/D electrode structure, the active area of each of the fully stretchable transistor needs to be larger than 3.5 × 5 mm to provide a $W/L$ ratio greater than 140, thus permitting a sufficiently high drain

current (>10$^{-4}$ A) to turn-on the individual OLEC pixel. Hence, our first goal is to achieve uniform patterning of a large-area fully stretchable transistor array with thin gate dielectric layer, high yield of transistors and minimal leakage current. The lack of high-quality and low-leakage thin elastic dielectric materials is one of the major limiting factors for a large-area-uniform, solution-processed patterning of fully stretchable transistors. Previous work reported polystyrene-*block*-poly(ethylene-*ran*-butylene)-*block*-polystyrene (SEBS) or polydimethylsiloxane (PDMS)[14,15,27] as the dielectric layer for stretchable carbon nanotubes- or organic-based TFTs. However, most elastomers have poor chemical resistance and are easily swollen by solvents used for subsequent processing, which result in a high leakage current. While elastic poly(vinylidene fluoride-co-hexafluoropropylene) (PVDF-HFP)[28,29] has a relatively high dielectric constant and good organic solvent resistance, it is susceptible to unintentional ionic impurities[30] and can readily result in the double-layer capacitive effect, thus hindering its application to drive the LEC array to enable fast on-off cycles necessary for display applications.

To address these limitations, we employed perfluorinated elastomers[31,32] for stretchable OTFT array fabrication. Perfluorinated elastomers are both stretchable and chemically inert towards organic solvents typically used for patterning solution-processed polymer semiconductors and conductors. We envision that using a strong solvent-resistant perfluorinated elastomer as stretchable gate dielectrics will allow us to directly print stretchable semiconductors in a large-area uniform manner. In addition, to address issues with the low adhesion between the perfluorinated surface with the organic semiconductor in order to avoid delamination during stretching, we introduce a previously reported crosslinker containing flexible polydimethylsiloxane that can not only enhance the stretchability of the semiconductor polymer but also crosslink the semiconducting and gate dielectric layers to provide an improved interfacial bonding (Fig. 1a)[33].

The monomers of our perfluorinated elastomer were synthesized by modifying the hydroxyl groups of perfluoropolyether (PFPE) diols with two methacrylate (DMA) groups[31] (Fig. 1b and Supplementary Fig. 1). The molecular weight of the initial PFPE diols can be used to tune the stretchability of the elastomer and effectively adjust the crosslinking density of the final elastomer. For example, a 12 kg/mol PFPE-DMA monomer yields a crosslinked elastomer with a fracture strain up to 200% while the 4 kg/mol PFPE-DMA monomer results in a fracture strain of ca. 10% (Supplementary Fig. 2, ref. [30]). In order to gauge the chemical resistance of the crosslinked PFPE-DMA film relative to other conventional stretchable dielectric polymer films (e.g. SEBS and crosslinked PDMS), we investigated the thin-film roughness prior and post solvent treatment with profilometry. We compared films with the same thickness treated by solvents typically used for dissolving polymer semiconductors (e.g. chloroform, 1,1,2-trichloroethene and tetralin). As expected, PDMS and SEBS films showed substantial non-uniformity in thickness after solvent treatment, an indication of substantial swelling or even partial dissolution (Fig. 1c, d). In contrast, PFPE-DMA films remained smooth and uniform after solvent treatment, consistent with our hypothesis. This solvent resistance of crosslinked PFPE-DMA is the key to enable the following fabrication process of our transistors.

The PFPE-DMA films showed a high-breakdown voltage of >100 V even though the thickness of the film was less than 200 nm (Supplementary Fig. 3) demonstrating its applicability as a gate dielectric for OTFTs. The dielectric constant of PFPE-DMA was measured to be 2.4 ± 0.1 at 20 Hz (Supplementary Fig. 4). By co-polymerization of PFPE-DMA with other fluorinated methacrylate monomer, such as 2-perfluorohexylethyl acrylate and pentafluorophenyl acrylate, the dielectric constant can be

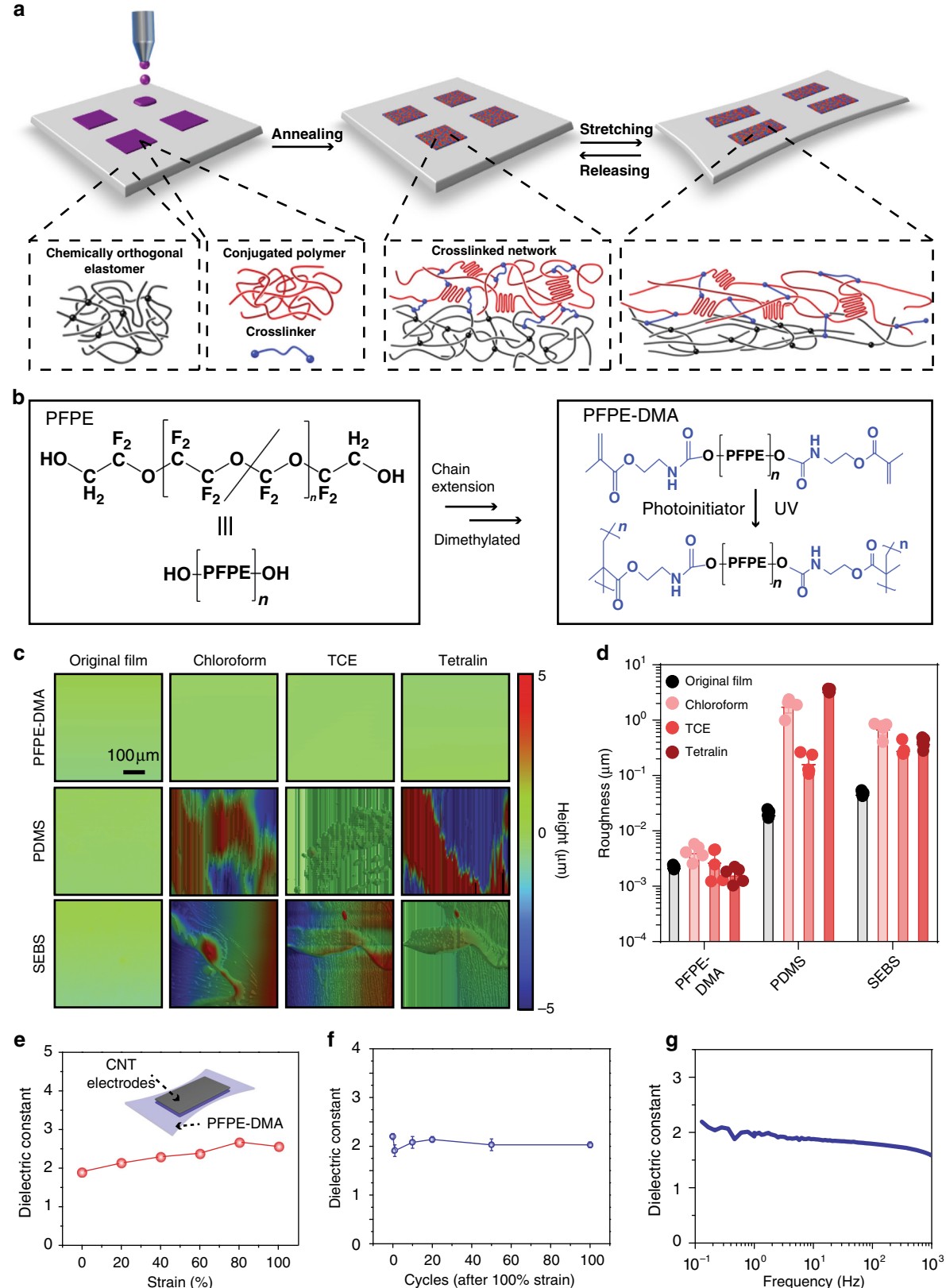

substantially enhanced to ca. 5.0 ± 0.2 at 20 Hz without affecting the macroscopic chemical orthogonality (Supplementary Fig. 4). Notably, the uniformity of the PFPE-DMA/2-perfluorohexylethyl acrylate and PFPE-DMA/pentafluorophenyl acrylate thin films need to be further improved for fully stretchable TFT array fabrication. To investigate the changes in dielectric constant of

PFPE-DMA film under different uniaxial strains, we fabricated a simple parallel plate capacitor with stretchable carbon nanotubes (CNT) electrodes as the conductive plates and crosslinked PFPE-DMA as the dielectric. Considering the geometrical change, less than 10% of dielectric constant change was observed from 0 to 100% strain (Fig. 1e) and the dielectric constant was stable after

**Fig. 1 Perfluorinated elastomeric gate dielectric layers. a** Schematics show the materials design for patterning of stretchable transistors on elastomeric dielectrics by inkjet printing. From left to right: First, semiconducting conjugated polymer with crosslinkers are printed on an ultra-thin elastomer film, which is resistive to that solvents used for semiconducting polymers. Second, crosslinkers crosslink the conjugated semiconducting polymers under thermal anealing, enabling their stretchability. The crosslinkers, at the same time, crosslink conjugated polymers with elastomeric dielectric layer, allowing for a tight bonding between semiconducting and dielectric layers under stretching. **b** Schematic of molecular structures illustrating the synthesis and photo-crosslinking of dimethacrylate-modified perfluoropolyether (PFPE-DMA) elastomer. **c** Representative 2D profilometer scanning images showing the roughness of PFPE-DMA and conventional stretchable dielectric thin films (5-μm thick) prior and post to the 1-min treatment of organic solvents, which are used for the solution-processed fabrication of stretchable organic thin-film transistors (OTFTs). TCE: 1,1,2-trichloroethene. Polydimethylsiloxane (PDMS) is standard Sylgard 184 crosslinked PDMS with 10:1 weight ratio of PDMS and crosslinker. The polystyrene-*block*-poly(ethylene-*ran*-butylene)-*block*-polystyrene (SEBS) compounds H1221, with poly(ethylene-co-butylene) volume fractions of 88% was provided by the Asahi Kasei company without crosslinking. **d** Statistical analysis of surface roughness changes of stretchable dielectric thin films after the solvent treatment ($n = 5$). **e** The dielectric constant characterization of PFPE-DMA thin film subjected to uniaxial strains. Inset illustrates the geometry of the freestanding device containing spray-coated carbon nanotube (CNT) stretchable electrodes that sandwich PFPE-DMA film for the dielectric characterization. **f** Cyclic stability of the dielectric performance of the PFPE-DMA thin film subjected to 100% strain. Values are mean ± S.D. **g** Dielectric performance of thin-film PFPE-DMA at frequency ranging from 0.1 to $10^3$ Hz.

100 cycles of 100% strain (Fig. 1f). Moreover, negligible changes in dielectric constant was observed for a frequency range of 0.1–$10^3$ Hz (Fig. 1g) Collectively, our results indicate that there was a minimal ionic effect in our PFPE-DMA films and suggest that PFPE-DMA could be used as elastic gate dielectrics in fully stretchable transistors.

To control the quality of spin-coated polymer semiconductor for characterizing the dielectric performance of PFPE-DMA, we fabricated a top-gate-bottom-contact (TGBC) OFET to evaluate PFPE-DMA as a suitable stretchable dielectric material (Fig. 2a). Notably, due to the chemical orthogonality of PFPE-DMA and fluorinated solvents (1, 1, 1, 3, 3-pentafluorobutane and trifluorotoluene), we were able to directly fabricate device in the TGBC configuration for both *p*-type and *n*-type organic semiconductors, showing typical current-voltage transfer characteristics (Fig. 2b and Supplementary Fig. 5). Specifically, for a 1.5-μm thick dielectric layer, the *p*-type OFET showed a saturation mobility of $0.43 \pm 0.1$ cm$^2$V$^{-1}$s$^{-1}$ and an on/off ratio of around $10^3$. According to previous reports[34,35], the carrier mobilities of semiconducting polymers are strongly dependent on the carrier density; a higher carrier density generally results in increased trap-filling and a more smoothing of electrostatic potential variations in the polymer film. These combined effects may lead to higher carrier mobilities. By reducing the thickness of the dielectrics, the vertical electric field will increase and thus increase the mobility measured in the horizontal direction. Upon reducing the thickness of the dielectric layer while measuring at the same applied gate voltage, the mobility of our thin-film transistor increases as expected since the mobility of organic semiconductor thin film is typically electric-field dependent (Fig. 2c) as opposite to the behaviour of TFTs made of ionic dielectrics, which do not show much thickness dependence[30,36,37]. Together, these results further reinforce our previous conclusion that there are only minimal ionic effects in the PFPE-DMA dielectric material, and it is potentially suitable for fast-switching OFETs.

**Inkjet printing of OTFTs directly on PFPE-DMA thin films.** Given the ccapability of inkjet printing to pattern organic semiconductors over a large area[38,39], we used inkjet printing to pattern the intrinsically stretchable semiconductor directly on PFPE-DMA as the dielectric layer to prepare the fully stretchable OTFT array. To achieve uniform patterning of the semiconductor, the polymer semiconductor must exhibit high solubility in the organic solvents and possess good ambient stability as the printing is typically conducted at ambient environment. Moreover, the dielectric surface needs to have suitable wettability. To meet the aforementioned requirements, we first developed a polymer semiconductor ink based on the isoindigo-containing semiconductor (PII2T)[40] mixed with an azide functionalized PDMS crosslinker (Supplementary Fig. 6)[39] for inkjet printing (Fig. 2d). This design allows for a high initial solubility of a non-crosslinked stretchable semiconductor precursor in tetralin solvent. After printing, the thermal treatment used to anneal the semiconductor simultaneously crosslinks the PII2T polymer with the PDMS-azide crosslinker into a stretchable network (Fig. 2d)[33]. Notably, without the PDMS-azide crosslinker, PII2T can only be stretched to ca. 15% strain before micron-sized cracks in the thin film appear. As calculated from nanoindentation characterizations, the modulus of the polymer semiconductor thin film crosslinked by PDMS-azide is reduced by ~2–3 times comparing to its pristine form (Fig. 2e, f). Further, the modulus stabilized when the weight ratio was increased from 100 to 200 wt%. Next, we investigated the OFET saturation mobilities of the crosslinked semiconductors with different weight ratios of crosslinker, where trimethoxy (octadecyl)silane (OTS) modified silicon oxide is the dielectric layer and thermally evaporated gold (40 nm) electrodes are the S/D top contacts (Fig. 2g). Even though the mobilities of the transistors decreased with addition of crosslinkers, they maintained similar values when the concentration of the crosslinker varied between 50 and100 wt% (Fig. 2h). Therefore, considering the relative mechanical and electronic performances, an optimized ink composition of 1:1 weight ratio of PII2T and crosslinker was selected.

Due to the high hydrophobicity of the perfluorinated material, a major challenge is direct inkjet printing of uniform semiconducting polymer patterns with high yield on the pristine PFPE-DMA film. Thus, to enhance the surface wettability, we modified the surface of PFPE-DMA with a thin layer of the PDMS-azide crosslinker (Fig. 2i). This treatment resulted in a significant change in water contact angle from 120° to below 30° (Fig. 2j). Consistent with the improved wetting, the inkjet-printed single droplet of ink (1:1 weight ratio of PII2T:crosslinker) on the surface of PFPE-DMA increased from a 10 μm area (shrunk droplet) to an 80 μm area (fully spread-out) after treatment (Supplementary Fig. 7a, b). Thus, PII2T polymer semiconductor patterns can be uniformly printed on millimetre length scales (Supplementary Fig. 7c, d, Fig. 2k). The modified surface also allows CNT electrodes to be uniformly spray-coated with no swelling to the bottom dielectric layers (Fig. 2l). After the thermal annealing process, the azide crosslinker in the ink reacts both with the semiconductor polymer as well as the gate dielectric layer, reducing the potential for delamination between the semiconductor and gate dielectric, which typically correlates with mechanical deformation. Indeed, the printed polymer semiconductor with improved layer adhesion did not show any wrinkles

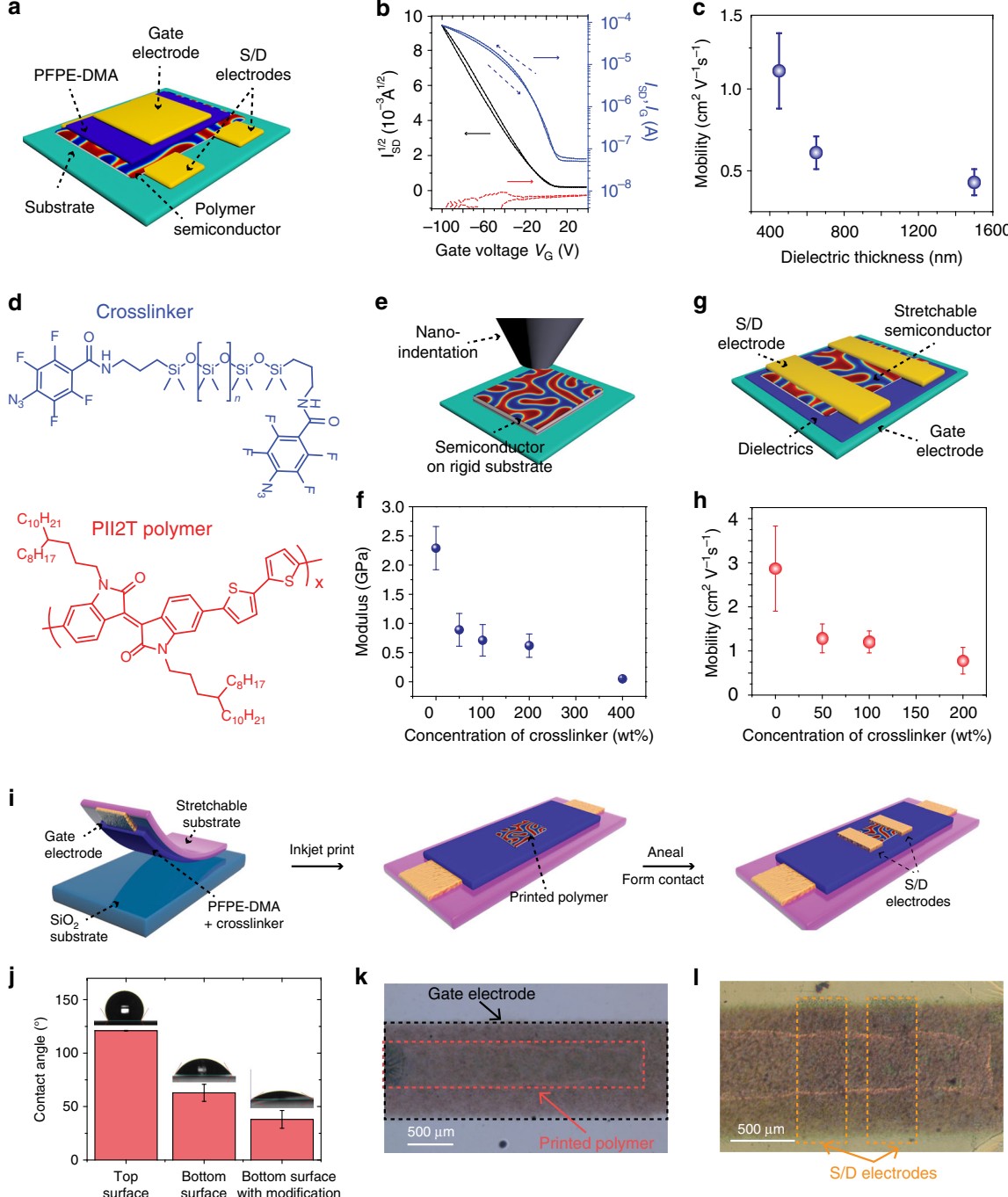

or cracks after 100 cycles at 100% strain as observed by bright-field imaging (Supplementary Fig. 7e–g).

**Fully stretchable OTFT array**. We designed a fabrication process to incorporate stretchable electronic materials into the stretchable transistors array as shown in Fig. 3a. Notably, PII2T polymer semiconductor patterns can be uniformly printed on millimetre length scales (e.g. $1.4 \times 1.2$ mm patterns). S/D electrodes were patterned by spray coating after inkjet printing of the semiconductor (Fig. 3b) to make a $5 \times 5$ transistor array (Fig. 3c). The fully stretchable active matrix can be stretched up to 100% (Fig. 3d). The measured average saturation mobility is $0.56 \pm 0.17$ $cm^2$/Vs (Figs. 3e, f, and Supplementary Fig. 8), which can be stretched up to 100% and cycled up to 1000 times with less than one order of magnitude reduction in both the mobility and on-

current (Fig. 3g, h), and minimal changes in leakage current (Supplementary Fig. 9).

**Vertically integrated AMOLEC array**. Figure 4a shows the device structure of a single-pixel OLEC. To prepare a stretchable OLEC, we used polyurethane acrylate coated by Ag nanowires (PUA-AgNWs) as the electrodes and PEDOT:PSS as the hole injection layer to sandwich the stretchable light-emitting layer (Fig. 4a, top). Here the PEDOT:PSS layer can effectively decrease the leakage current, enhancing the device performance. The light emission layer contains a blend of a light-emitting polymer (Super Yellow, SY), ion-conducting polymer, ethoxylated trimethylopropane triacrylate (ETT-15) and lithium trifluoromethane sulphonate (LiTf) with the weight ratio of 20:20:2:1. Here we used the ion-conducting polymer[41]

**Fig. 2 Fully stretchable thin-film OTFTs. a** Schematic of the top-gate-bottom-contact (TGBC) structure of isoindigo-based PII2T polymer using the PFPE-DMA as the gate dielectric (thickness 500 nm to 1.5 μm and dielectric constant of ca. 2.2), Al/Au as gate electrode and Cr/Au as the source and drain (S/D) electrodes. $W/L = 20$, $V_{sd} = 100$. **b** Representative transfer characterization of the TGBC structure OTFTs. **c** The mobility value of OTFTs to the thickness of PFPE-DMA gate dielectrics. **d** Chemical structures of (PDMS)-azide crosslinker and PII2T semiconductor polymer as stretchable semiconductor ink for inkjet printing. The molecular weight of PDMS is 3100 g/mol. **e** Schematic of nanoindentation used for the Young's modulus measurement. **f** Young's modulus of the intrinsically stretchable semiconductor polymer to the concentration of the PDMS-azide crosslinker in PII2T polymer. **g** Schematic of the top-contact-bottom-gate (TCBG) device structure for measuring values of mobility. **h** The saturation mobility of OTFT as a function of the concentration of PDMS-azide crosslinker in PII2T polymer. The dielectric is octadecyltrichlorosilane (OTS) modified $SiO_2$, the drain and source electrodes are thermally evaporated gold (40 nm) with $W/L = 20$. **i** Schematics of the stepwise process for the fabrication of a freestanding, fully stretchable OTFT array through inkjet printing. The gate electrode was prepared by spray using CNT/ Poly(3-hexylthiophene) (P3HT) with a thickness ~5 μm. The drain and source electrodes were prepared by spray coating using CNT/P3HT material with a thickness of ~10 μm. The $W/L = 2$. The PFPE dielectric layer thickness is 1.5 μm with a capacitance per area of 1.4 nF/cm². **j** Statistical analysis of contact angle measurement for PFPE-DMA film with surface different modifications. Inset images show representative contact angle measurements for each condition. Top surface: top surface of PFPE-DMA film spin-coated on dextran/$SiO_2$ substrate. Bottom surface: bottom surface of PFPE-DMA film spin-coated on dextran/$SiO_2$ substrate. Bottom surface with modification: bottom surface of PFPE-DMA spin-coated on PDMS-azide/dextran/$SiO_2$ substrate post-baked at 150 °C for 40. Values are mean ± S.D. **k** Optical bright-field (BF) image shows inkjet-printed semiconducting polymers on the PFPE-DMA gate dielectric layer with the bottom-gate electrode. **l** Optical BF image shows the thermal annealed semiconducting polymers patterned with top S/D CNT stretchable electrodes.

synthesized by our group instead of the commonly used poly-ethylene oxide (PEO) is mainly because of its higher stretchability than PEO. ETT-15 serves as the ionically conductive component that can be polymerized by heating, and LiTf can provide the ionic dopant to form the PIN junction in the emission layer. The resulting OLEC can be stretched up to 30% strain without delamination or crack formation (Fig. 4b). To balance the stretchability while maintaining sufficient light-emitting cell current density, the optimized weight ratio of SY to ionic conducting polymer was found to be about 1:1. The turn-on-current density of the LEC is around 2 mA/cm² (Fig. 4c). The luminance ratio of the OLEC is increased with the applied voltage (Supplementary Fig. 10). Importantly, the current density from a single OLEC device was stable when subjected to less than 30% strain (Fig. 4c). The working mechanism of the OLEC is described below. When a voltage is applied to an OLEC, ions in the emission layer are redistributed to form an electrical-double layer at anode (PEDOT: PSS/PUA/AgNW) and cathode (PUA/AgNW) interfaces to allow easy hole or electron injection, respectively, leading to electro-chemical doping to form a light-emitting PIN junction. The Supplementary Table 1 provides a summary of some related research works on PLEC with excellent flexibility and stretchability.

Next, the individual pixel of the OLEC were integrated with each drain electrode on the stretchable active matrix through via filled with silver conductors (Fig. 4d). The stretchable polymer semiconductor was printed as described earlier with an area of 0.8 × 0.5 cm. The interdigitated S/D electrodes have a $W/L$ ratio of 140. The PFPE-DMA substrate was slightly treated by oxygen plasma to dope the semiconductor in order to obtain sufficient current density to drive the LEC while its off-current is still low enough to keep the OLEC properly "off" (Fig. 4e, f). Both photographic image and video (Fig. 4g and Supplementary Movie 1) illustrate the control of the OLEC pixels by the stretchable OTFT underneath. Both the on-current and current density of the transistors showed minimal changes when subjected up to 30% strain (Fig. 4e), and they were capable of maintaining the light illuminating of the OLEC (Fig. 4h).

Finally, we vertically integrated the fully stretchable active matrix with the OLEC array to complete our AMOLEC array (Fig. 4i). The average drain-current was 0.16 mA ± 0.04 mA from a 5 × 5 OTFT array occupying 3.9 × 3.7 cm² (Fig. 4j). Due to the low moduli materials used for our stretchable skin display structure, our obtained AMOLEC array possesses mechanical softness (~50 MPa, Supplementary Fig. 11). We observed that even after being subjected to 20 cycles at 30% strain, the resulting

cross-sectional image obtained via scanning electron microscopic (SEM) showed negligible delamination in the multilayered structure (Fig. 4k). In the 3D vertically integrated structure of our skin display device (Supplementary Fig. 12a), each active layer was confirmed using energy-dispersive X-ray (EDX) mapping (Supplementary Fig. 12b), where we can clearly identify the distinct elements attributed to each active layer (e.g. F-element for the PFPE-DMA dielectrics and Ag for the stretchable transparent electrodes). Finally, Fig. 4l shows a freestanding 2 × 3-pixel array in its "on"-state at 0% strain while the pixels in the display can be controlled using OTFTs. Figure 4m further illustrates the array subjected to different bending, twisting and stretching states while all the pixels are 'turned-on'.

## Discussion

We present herein the first demonstration of a stretchable active-matrix-driven OLEC array (termed 'skin display' device) that possesses intrinsic and full stretchability like human skin. Our skin display devices were fabricated via all-solution processing of intrinsically stretchable materials, rendering these processes potentially scalable. Furthermore, with its excellent intrinsic stretchability, our skin display devices can be potentially vertically integrated with other fully stretchable sensor arrays to provide human-interactive systems for sensing on-body information display and detection through visual interaction. Building on this work, and in conjunction with the rapid progress in the field of stretchable transistor array and light-emitting device development, we envision the density of pixels can be further increased by utilizing transistors arrays with higher mobilities. Further optimization of the fluorinated polymers with higher dielectric constant should further reduce the S/D operating voltage needed to facilitate the integration of our fully stretchable skin display with human body.

## Methods

**Materials**. *Synthesis of perfluorinated elastomer as a dielectric layer for OTFT:* Supplementary Fig. 1 shows the synthesis of dimethacrylate-functionalized per-fluoropolyether monomer (PFPE-DMA). This synthesis process follows a previous report[28]. Briefly, the PFPE diols (4 kg/mol) was obtained from Solvay Specialty Polymer. It was first dissolved in 1,1,1,3,3-pentafluorobutane (Alfa Aesar, 406-58-6) and reacted at room temperature with isophorone diisocyanate (IPDI, Sigma–Aldrich, #317624) at a 2:1 molar ratio for 48 h to yield the chain extended PFPE diols. Then, the product was reacted with 2-isocyanatoethyl methacrylate (IEM, Sigma–Aldrich, #477060) at a 1:1 molar ratio (4 kg/mol PFPE diols:IEM) at room temperature for 48 h. In both reactions, 0.1 wt% Tributyltin acetate (DBTDA, Sigma–Aldrich, #347787) was used as the catalyst. The final product was filtered through a 0.2 μm syringe filter to yield a clear and colorless oil. The solvent was removed by rotary evaporation. Afterwards, PFPE-DMA solution was diluted in

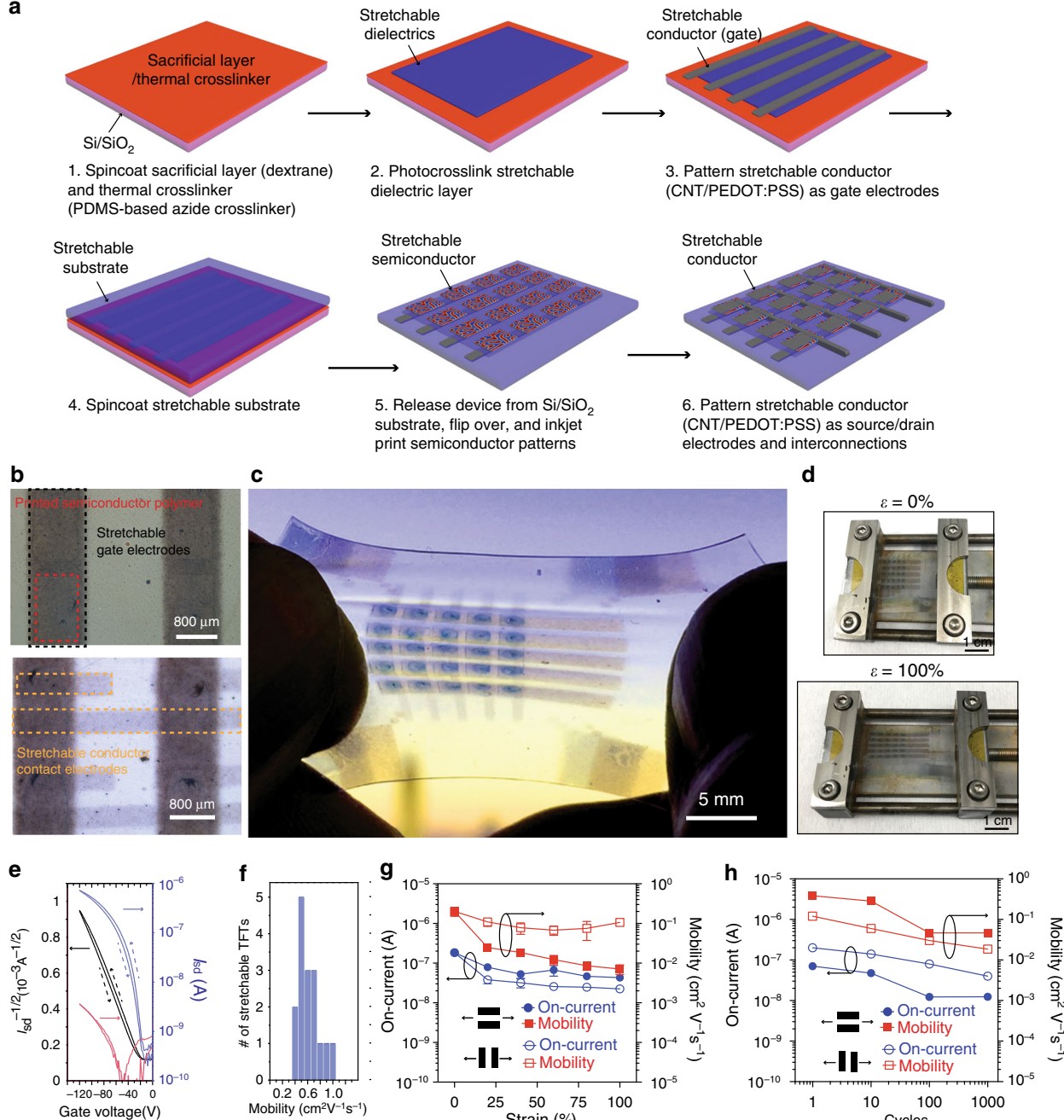

**Fig. 3 Fully stretchable OTFT array. a** Fabrication process. A water-soluble sacrificial layer was spin-coated on a Si/SiO$_2$ wafer followed by a PDMS-azide layer. A chemically orthogonal stretchable dielectric layer (PFPE-DMA) is spin-coated and UV cured. A stretchable conductor as the gate electrode is spray-coated through a shadow mask first with a mixture of CNT/poly(3,4-ethylenedioxythiophene) flowed by a PEDOT:PSS layer. Then a stretchable substrate made of PDMS is spin-coated. After curing, the entire structure is soaked in water and is released from the substrate and flipped over. Inkjet printing is used to pattern the stretchable semiconductor on the stretchable dielectric layer. Stretchable electrodes made of CNT/PEDOT:PSS are spray-coated through a shadow mask to give the source/drain (S/D) electrodes. **b** Optical microscopic images of a printed stretchable semiconductor pattern (red dashed box) on the PFPE-DMA dielectric film with stretchable CNT gate electrodes (black dashed box)and a fully stretchable transistors array with top CNT/PEDOT:PSS stretchable S/D contact electrodes (yellow dashed boxes) on the patterned semiconductors. **c** An optical photographic image of a fully stretchable 5 × 5 transistor active-matrix array. **d** Optical photographic images show the stretchable transistor array being subjected to 0% and 100% strain. **e** A representative transfer curve ($V_d = -100$ V) of the stretchable OTFT at 0% strain. **f** Distribution of the mobility from 16 devices on the fully stretchable OTFT array (Supplementary Fig. 8). **g** Changes in the saturation on-current and saturation mobility at gate voltage of −100 V and drain source voltage of −100 V with strains up to 100%, both parallel to (open circles) and perpendicular to (filled circles) the charge transport direction ($n = 3$). Values are mean ± S.D. **h** Changes in the saturation on-current and saturation mobility measured at gate voltage of −100 V and drain source voltage of −100 V after multiple stretching-releasing cycles (up to 1000 cycles) at 100% strain parallel to (open circles) and perpendicular to (filled circles) the charge transport direction.

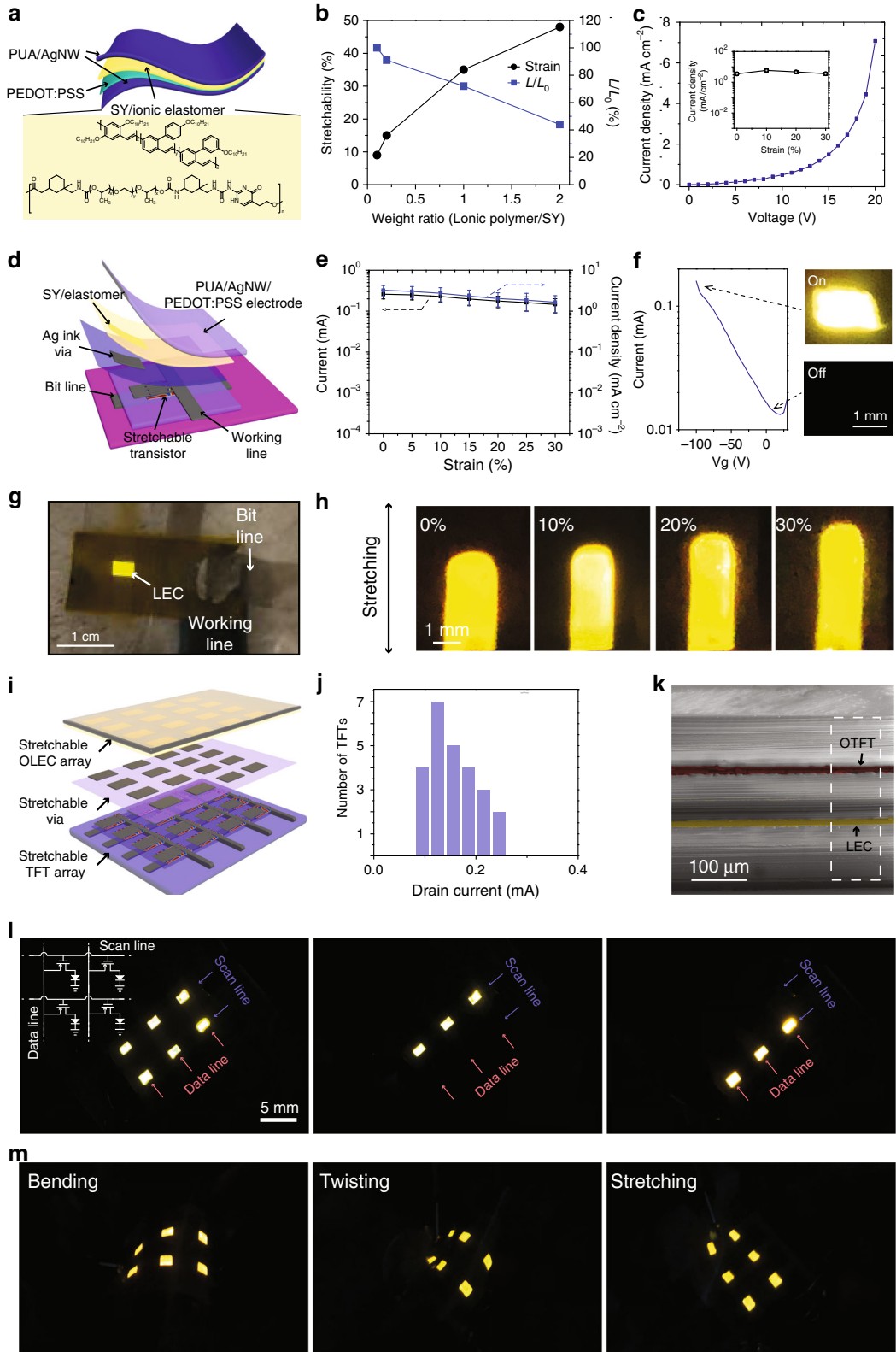

1,3-bis(trifluoromethyl)benzene (weight ratio 1:2). 0.3 wt% of 2-hydroxy-2-methylpropiophenone was added as a photo initiator. The mixed solution was filtered through a 0.45-μm filter.

*Stretchable semiconductor ink for inkjet printing*: The ink for the inkjet printing was prepared by dissolving polyisoindigo-based polymer (PII2T, $M_n$ = 51 kg/mol, $M_w$ = 138 kg/mol, D = 2.7) and the PDMS-azide crosslinker reported previously[34] in 1,2,3,4-tetrahydronaphthalene (Tetralin, Sigma–Aldrich, #522651) at 1:1 weight ratio with a concentration of 2.5 mg/mL. The dissolved ink was then filtered through a 0.7-μm filter. The $M_n$ of the PDMS was 3.1 kg/mol.

*Precursor solution for stretchable light-emitting electrochemical cell (LEC)*: SuperYellow (phenyl substituted poly(1,4-phenylene vinylene)) was provided by Merck. The stretchable ion-conducting polymer was synthesized using a previously reported method from our group[35]. Lithium trifluoromethanesulfonate (LiTf, 99.995% purity), ethoxylated trimethylolpropanetriacrylate and anhydrous tetrahydrofuran (THF) were received from Sigma–Aldrich and used as is. SuperYellow, ion-conducting polymer, ethoxylated trimethylolpropanetriacrylate and LiTf in THF (weight ratio 20:20:2:1) with 7 mg/mL of Super Yellow were prepared in glovebox.

**Fig. 4 Fully stretchable AMOLEC array. a** Schematic of the stretchable light-emitting electrochemical cell (LEC). Yellow highlighted schematics show the molecular structure of stretchable ion-conducting polymer and super yellow (SY) used to enable the stretchable, tissue-level soft LEC. **b** Strain tolerance and light-emitting intensity ratio (light-emitting intensity/initial light-emitting intensity, $L/L_0$) of LEC to the different weight ratios of ionic polymer to SY. **c** Representative curve of current density as a function of the voltage of the stretchable LEC with 1:1 weight ratio of stretchable polymer: SY. It shows a stable current density as a function of strain (inset). The active area of the device is $2 \times 3$ mm$^2$. The "on" state of the LEC is defined as current density above 2 mA/cm$^2$ as chosen by curves in **c**. **d** Schematic 3D layout of individual stretchable LEC pixel driven by a stretchable OTFT. **e** Drain current and the corresponding current density of the fully stretchable OTFT subjected to strains from 0 to 30%. Values are mean ± S.D. **f** The on-off-current from the transistor controlled by the gate voltage and the representative images of LEC at "off" and "on" states. **g** Photographic image of a stretchable LEC pixel turned-on by the vertically integrated stretchable transistor. **h** Photographic images of a single LEC pixel subjected to different strains. **i** Schematic of the vertical integrated active matrix and LEC array. **j** The histogram of drain-currents from a typical stretchable OTFT active matrix. **k** A SEM cross-sectional image of an AMOLEC device stack after subjected to 30% strain and released. The false colour highlights the active layer of OTFT (red) and LEC (yellow), respectively. **l** Photographic images of the AMOLEC skin display pixels with different columns being selectively turned on. **m** Photographic images of AMOLEC skin display pixels subjected to both bending, twisting and stretching.

**Fabrication**. *Fabrication of top-gate-bottom-contact transistor for dielectric performance testing*: First, Cr/Au (5 nm/100 nm) electrodes were evaporated on a glass substrate through a shadow mask. After 1-min oxygen plasma treatment (Technicx MICRO-RIE SERIES 800, O$_2$ pressure 200-ppm, power 150 W), a solution of PII2T or FBDPPV[42] in trichloroethylene (TCE, Sigma–Aldrich, #79-01-6) (1 mg/mL) was spin-coated at 1000 r.p.m followed by baking at 120 °C for 1 h. Next, a PFPE-DMA solution (1:2 weight ratio in 1,3-bis(trifluoromethyl)benzene) was directly spin-coated on the semiconductor layer, followed by 15-min UV curing (365 nm) and 1-hour baking at 120 °C in a N$_2$-filled glovebox. Finally, Al/Au (50 nm/50 nm) top-gate electrodes were evaporated on top of PFPE-DMA through a shadow mask.

*Fabrication of bottom-gate-top-contact transistors for stretchable semiconductor ink testing*: Semiconductor inks with different concentrations were spin-coated on the top of the octadecyltrichlorosilane (OTS) modified silicon oxide wafer as previously reported[14]. Au (50 nm) electrodes were evaporated on top of the semiconductor layer as source-drain electrodes (S/D) through a shadow mask.

*Fabrication of the intrinsically stretchable transistors array*: First, the bare silicon wafer was cleaned using deionized (DI) water/isopropyl alcohol (IPA) in an ultrasonic bath for 10 min, dried by N$_2$ gas and then was oxygen plasma (Technicx MICRO-RIE SERIES 800 O$_2$ pressure 200ppm, power 150 W) treated for 30 s. Second, dextran solution (5 wt% in water) was spin-coated on the silicon wafer at 2000 r.p.m. for 60 s. The dextran thin film was baked at 100 °C for 5 min to remove water. Third, a solution of PDMS-azide crosslinker (15 mg/mL in IPA) was spin-coated on top of dextran layer at 3000 r.p.m. for 60 s, followed by a soft baking at 40 °C for 5 min to remove the extra solvents. Fourth, PFPE-DMA solution was spin-coated at 2000 r.p.m. for 60 s on the modified dextran layer as dielectrics. The spin-coated PFPE-DMA film was cured by a 15-min UV curing (365 nm, 1 h) and post-baked at 150 °C for 40 mins in glovebox. Fifth, the PFPE-DMA film was oxygen plasma (O$_2$ pressure 200ppm, power 150 W) treated for 30 s. A single-wall carbon nanotube (SW-CNT, Carbon Solutions, Inc. P2-SWNT #02-A013) solution (12 mg/70 mL in chloroform, sonication for 20 mins and centrifuge for 30 mins) was spray-coated with a shadow mask as the bottom-gate electrode. Sixth, PDMS (Sylguard 184, 10:1 PDMS:crosslink ratio) was then spin-coated at 700 r.p.m. for 60 s as a bottom encapsulation layer and was post-baked at 70 °C for 2 h. Seventh, this multiple layered structure was then lifted off by immersing in DI water for 1 h to dissolve the underneath dextran layer. Eighth, the top layer of PFPE-DMA was then oxygen plasma (O$_2$ pressure 200ppm, power 150 W) treated for 3 min to increase wettability before the inkjet printing. Ninth, the stretchable semiconductor ink solution as described in section 1.2 was then inkjet-printed as the semiconducting layer. Inkjet printing was done using a Dimatix Materials Printer (DMP-2850). The drop-to-drop distance was set to 50 μm. The temperature of the sample substrate was set to 40 °C. Three to four layers of consecutive printing were used to define the semiconducting polymer patterns on the substrate. The printed patterns were baked at 120 °C for 1 h. Finally, a thin PEDOT:PSS layer (Heraeus CLEVIOS$^{TM}$ PH 1000 1:5 volume ratio diluted in methanol) was spray-coated through shadow mask as a protective layer, and then SW-CNT solution was spray-coated through a shadow mask as source-drain electrodes. The device was post-baked at 100 °C for 1 h in the glovebox.

*Fabrication of freestanding LEC*: First, 1 mg/mL AgNW/IPA solution (Zhejiang Kechuang Advanced Materials Technology Co., Ltd.) was spray-coated onto a glass substrate. Second, polyurethane (PUA) precursor contains urethane (UA, Sartomer, CN990), dimethacrylate (EBA, Sartomer, SR540) and DMPA (Sigma–Aldrich #196118) at a weight ratio of 100:20:1 was drop-casted on top. The coating was UV cured (365 nm) for 10 min and was then peeled off as the freestanding electrode. Third, a PEDOT:PSS (Heraeus CLEVIOS$^{TM}$ PH 1000) layer was spin-coated on the anode electrode at 1000 r.p.m. for 60 s as a protective layer and was dried in vacuum for 4 h. Fourth, the SuperYellow (SuperYellow, ion-conducting polymer, ethoxylated trimethylolpropanetriacrylate and LiTf in THF with a weight ratio of 20:20:2:1 and a concentration of 7 mg/mL SuperYellow) was spin-coated at 1000 r.p.m. for 60 s in glovebox as the light-emitting layer, and

was dried in vacuum for 1 h. Finally, a second PUA/AgNW electrode layer was then manually stacked on the light-emitting layer as the cathode. The device was post-baked at 90 °C for 30 min to increase layers adhesion.

*Fabrication of fully stretchable skin display*: The AgNW/PU electrode was first prepared based on the abovementioned method. PEDOT:PSS was then spin-coated onto it at 1000 r.p.m. and dried in vacuum for 4 h, which was then used as the anode. A solution of electroluminescent polymer layer (the SuperYellow solution mentioned above) was spin-coated onto the anode at 1000 r.p.m. for 60 s, which was vacuum dried for 1 h. A patterned AgNW/PU electrode (as cathode) prepared (same procedure as above, $2 \times 3$ pixels and a $2 \times 3$ mm dimension of each pixel) using the shadow mask was faced down onto the electroluminescent polymer layer. The resulting array stack was heated to 90 °C to enhance adhesion along with hot pressing for several times.

**Characterization**. *Surface roughness characterization section*: PFPE-DMA solution was spin-coated on a silicon wafer at 2000 r.p.m. for 60 s. The films were then UV cured as described above in the glovebox for 15 mins, and baked at 150 °C for 40 min to be fully crosslinked. SEBS (H1052 from Asahi Kasei Corporation) was dissolve in toluene with a concentration of 80 mg/mL. The solution was filtered by through a 0.45-μm syringe filter, and was spin-coated on a silicon wafer at 2000 r.p.m. for 60 s. The films were then dried at room temperature for 2 h. PDMS (15:1 crosslink ratio) was diluted in toluene (weight ratio 1:3). The solution was filtered through a 0.45-μm syringe filter, and spin-coated on a silicon wafer at 3000 r.p.m. for 60 s. The films were baked at 70 °C for 2 h to be fully crosslinked. The films were 1-μm thick, as measured by a Filmetrics Profilometer (Filmetrics F20-EXR thin-film analyzer, 205-0593).

Trichloroethylene (TCE), 1,2,3,4-tetrahydronaphthalene (Tetralin) and chloroform (CF) solvent liquid (100 μL) were dropped on these prepared thin-film surface by pipette and let sit for 1 min. The solvent-treated thin films were then dried at room temperature for the following surface roughness characterization.

The thin films surface roughness characterization was performed by a Bruker Dektak XT surface profilometer. 3D surface mapping function was applied with a scale of $500 \times 500$ μm and with a resolution of 25 traces/μm.

*Dielectric characterization*: The dielectric performance as a function of frequency was acquired by a Bio-Logic VMP3 electrochemical workstation (>0.1 Hz). The strain and cyclic stability of the dielectric constant measurements were performed on the freestanding PFPE-DMA sandwiched by the spray-coated CNT electrodes ((SW-CNT, Carbon Solutions, Inc. P2-SWNT #02-A013) solution, 12 mg/70 mL in chloroform, sonication for 20 min and centrifuge for 30 min and acquired by an Agilent E4980A Precision LCR Meter.

*Organic thin-film transistor characterization*: OTFT characteristics were acquired using a Keithley 4200-SCS parameter analyzer under ambient atmosphere. The forward sweeps of the transfer curves were used for the mobility calculation. The reported mobilities are averaged values from at least ten devices for each condition, and the error bars denote the standard deviations.

*Modulus characterization*: Glass substrates were cleaned by IPA in an ultrasonic bath. PII2T isoindigo polymer was dissolved in TCE with a concentration of 2 mg/mL. PDMS-azide crosslinker was added into the solution with different weight ratio (azide crosslinker: PII2T as 0:1, 2:1 1:1, 1:2). The solutions were drop-casted on glass substrates and were dried at 40 °C. The dried films were post-baked at 100 °C to be fully crosslinked. The glass substrates were then mounted to a Aluminium puck using graphite paste. A cylindrical flat punch tip with a dynamic indentation method was used to probe the modulus of the films, and the measurement was performed on a Nanomechanics iNano Nanoindenter.

*SEM imaging characterization*: The stacked layers of the skin display were post-baked at 120 °C for 30 min and subjected to 30% strain. The device was then cut by Feather scalpel blade to expose cross-section. The cross-section characterization and EDS element mapping were performed by FEI XL30 Sirion SEM.

## Data availability
The data that support the findings of this study are available from the corresponding author upon reasonable request.

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

## Acknowledgements
We thank Tadanori Kurosawa, Simon Rondeau-Gagné and Yu-Cheng Chiu for their help on experiments. This work was supported by Samsung Electronics. G.J.N.W was supported by the Air Force Office of Scientific Research (grant no. FA9550-18-1-0143). B.C. S. acknowledges the National Research Fund of Luxembourg for financial support (Project No. 6932623). H.T. was supported by appointment to the Intelligence Community Postdoctoral Research Fellowship Program at Stanford University, administered by Oak Ridge Institute for Science and Education through an interagency agreement between the U.S. Department of Energy and the Office of the Director of National Intelligence. Y.L. thanks for support from National Natural Science Foundation of China (21420102005). Y.X.W. thanks the financial support from China Scholarship Council (201806255002). Part of this work was performed at the Stanford Nano Shared Facilities (SNSF), supported by the National Science Foundation under award ECCS-1542152. We thank Solvay Specialty Polymer for providing PFPE diols.

## Author contributions
J.L., J.W., Z.Z. and Z.B. designed the experiments. J.L., J.W. and Z.Z. performed the experiments. F.M.L. performed the initial testing on inkjet printing, G.N.W. synthesized PDMS-azide crosslinker. B.S. synthesized PII2T polymer. X.Y. synthesized ionic polymer. Y.Z. performed SEM and EDX characterization. O.Z. performed nanoindentation testing. T.L. and Y.L. synthesized the n-type semiconductor polymer. H.T. assisted with profilometry experiments. Y.X.W. performed NMR characterization. J.L., J.W., Z.Z. and Z.B. analysed the data. J.L., J.W., Z.Z., J.T. and Z.B. wrote the manuscript. All authors discussed the results and commented on the manuscript.

## Competing interests
The authors declare no competing interests.
