## [Peer Review File · Nature Communications]

Reviewers' Comments:

Reviewer #1:

Remarks to the Author:

Liu and Bao and coworkers present a study on the pioneering development of a stretchable active-matrix driven emissive display. The novelty of the approach and the potential impact of the results could definitely merit publication in a high-impact journal, such as Nature Communication. In this context, it is however unfortunate that the current presentation of the results is of an insufficient quality, which makes the reading and evaluation unnecessarily difficult. Thus, my recommendation is that the authors should perform a serious rewrite of the manuscript, and that an appropriately revised version should be provided with a second chance for publication. Below, I have included some guidelines for improvement, but wish to emphasize that the authors must take a holistic approach to their revision.

1. It is written that the OSC is "intrinsically stretchable" but on page 5 the authors state that "the originally brittle semiconductor" is endowed with stretchability by the small molecular crosslinker. The authors should clarify.
2. Page 6: it is stated that the dielectric constant of the gate dielectric can be improved to a value of 5 (from 2) with apparently facile means (which is obviously desired if the goal is to improve the drain current), but still they choose to not utilize this approach. Why?
3. (minor) Page 6: for the final device, the authors employ a bottom-gate configuration, but the initial evaluation of the dielectric is performed with a top-gate configuration. Why?
4. Paragraph spanning page 6-7: how can the mobility increase with decreasing dielectric thickness? To my understanding, the drain current will increase but not the mobility. The authors should explain how they anticipate that an increased vertical electric field will increase the mobility measured in the horizontal direction?
5. On the same topic, I do not understand how the claimed gate-dielectric thickness-dependence on the electronic mobility in the OSC can prove that ion motion in the dielectric is "minimal". My intuition tells me the opposite, and I recommend that the authors instead measure, e.g., the time dependence of the off current (which likely will change if ions can traverse the dielectric-OSC interface).
6. Page 7: it is stated that in order to pattern the OSC it is important that it features good ambient stability. Why?
7. On a number of occasions, the authors refer to the incorrect Figure, which must be corrected.
8. Fig. 2g: an explanation for the different measurement conditions should be included. For instance, what is the air interface?
9. Figs. 2i-j: the authors should clarify what the dashed lines indicate. On my screen and printout, the central grey region looks identical within and outside the dashed boundaries.
10. The LEC section is inadequate at this stage and must be rewritten. A few specific issues are described below.
11. The LEC is distinguished by the existence of mobile ions in the active material, and the selection and concentration of this critical element should be described in the main text (although I note that it is mentioned in the experimental section). Similarly, the main text suggest that it is only one ion conductor in the active material, while the experimental section reveals two. The authors must correct.
12. The turn-on time of LEC devices can vary from a few tenths of a second to several hours depending on the active material composition (and the drive voltage). This is obviously a very critical measure for the merit of an LEC in a display application, and the authors must include data on the temporal evolution of the current density and the luminance.
13. On a similar topic, as LECs are dynamic devices that change resistance during the in situ doping process, it is not meaningful to present data without disclosing under which conditions they were recorded (e.g. the time of recording should be included in Fig. 4c.)
14. It is also notable that quantitative information on the measured efficiency and luminance is lacking, and the authors should include this missing piece of information. It would also be very interesting to learn more about how their LECs perform in comparison to the state-of-the-art for stretchable and non-stretchable LECs. After all, essentially identical but non-stretchable LEC

devices void of the new ion transporter have been frequently reported.

15. Since the merit of a display obviously is (in part) defined by its peak light-emission intensity and pixel switching time, it would be interesting to be educated on the current status of the stretchable display as regards to these metrics. As I anticipate that parasitic capacitances and poor LEC performance could limit these metrics at this early stage of development, I emphasize that I am not expecting any record value, but more detailed information on the current performance would nevertheless be interesting.

Reviewer #2:

Remarks to the Author:

The authors claim that fully stretchable AM displays are important to skin electronics for wearables. The presented work demonstrates, to the authors knowledge, for the first time a fully stretchable AMOLEC display.

Although the work seems interesting, to my opinion, stretchable PMOLEC pixels have been demonstrated before in literature. In addition, OLED and AMOLED displays have shown to be fully flexible before, e.g. <https://doi.org/10.1002/jsid.547> by Samsung.

Therefore, to my opinion, the novelty of the work is not the first AM display in the world.

Moreover, the authors demonstrate the technology with only a 3x2 matrix. Regarding Fig 3g, can the authors explain why the mobility drops and increases again with increased strain? The TFTs seems to operate at high voltages, however the OLEC at 10V. Which are the voltages during operation? To my opinion, this would also limit the 'skin' operation.

The parameters of the OLED/TFT change after 10 cycles, I was wondering how many cycles are required for typical skin-electronics.

I would not recommend to publish this work in a Nature affiliated journal.

Reviewer #3:

Remarks to the Author:

The authors presented an intrinsically stretchable AM-OLEC array fabricated by solution-based inkjet printing. This is an extension work of the printed stretchable OTFT reported by the same group. I think the major improvement from the previous work is the studies on the dielectric layer and the fabrication of the active matrix LEC. Within my knowledge, this work is the first demonstration of the printed stretchable AM-OLEC display, which is a valuable advance in the printed electronics society. I recommend publication of this work after revising a few comments.

1. Detailed description on the transistor fabrication looks somewhat decreasing the importance of the stretchable dielectric layer for printing. I suggest, emphasize the dielectric part and the other transistor fabrication can be reduced.

2. In the dielectric layer, what is the minimum thickness in the aspect of the film stability and in the aspect of device operation (leakage). More details about the dielectric layer (such as capacitance, break-down voltage, thickness dependence, etc..) will be helpful to the readers.

Point-by-point response letter to reviewers' comments

We were delighted to see the strongly positive comments from all of the reviewers, e.g., “the novelty of the approach and the potential impact of the results could definitely merit publication in a high-impact journal, such as Nature Communication”, “the presented work demonstrates, to the authors knowledge, for the first time a fully stretchable AMOLEC display” and “this work is the first demonstration of the printed stretchable AM-OLEC display, which is a valuable advance in the printed electronics society”.

We were also grateful for the close read and helpful suggestions from all of the reviewers, and we have addressed in full every suggestion as detailed below. Thank you all again for your careful attention!

Reviewer #1

General Comment: Liu and Bao and coworkers present a study on the pioneering development of a stretchable active-matrix driven emissive display. The novelty of the approach and the potential impact of the results could definitely merit publication in a high-impact journal, such as Nature Communication. In this context, it is however unfortunate that the current presentation of the results is of an insufficient quality, which makes the reading and evaluation unnecessarily difficult. Thus, my recommendation is that the authors should perform a serious rewrite of the manuscript, and that an appropriately revised version should be provided with a second chance for publication.

Response: We thank the reviewer for the positive comments on our paper. We have revised the manuscript substantially to increase the readability of the paper and provide further clarification.

Comment-1: It is written that the OSC is “intrinsically stretchable” but on page 5 the authors state that “the originally brittle semiconductor” is endowed with stretchability by the small molecular crosslinker. The authors should clarify.

Response: We thank the reviewer for pointing out this confusion part. We want to clarify that the initial semiconductor polymers are also stretchable, which could be stretched up to 15% strain. We further included PDMS-azide crosslinker into the polymer network, where the PMDS provided flexible and stretchable linkers while inducing more disorders in the semiconducting polymer. After crosslinking with the semiconductor polymer backbone, these crosslinkers serve as stretchable components to further enhance the stretchability of the semiconductor polymers. The protocol and mechanism have been studied in detail in one of our previous publications (Wang, G.-J. N. *et al.* Inducing elasticity through oligo-siloxane crosslinks for intrinsically stretchable semiconducting polymers. *Adv. Funct. Mater.* **26**, 7254-7262, (2016)).

We have cited this reference as Ref #33. To further clarify this design, we have revised the manuscript as “.....to address issues with the low adhesion between the perfluorinated surface with the organic semiconductor in order to avoid delamination during stretching, we introduce a crosslinker containing flexible polydimethylsiloxane we reported previously that can not only enhance the stretchability of the semiconductor polymer but also crosslink the semiconducting and gate dielectric layers to provide an improved interfacial bonding (Fig. 1a)³³.”

Comment-2: Page 6: it is stated that the dielectric constant of the gate dielectric can be improved to a value of 5 (from 2) with apparently facile means (which is obviously desired if the goal is to improve the drain current), but still they choose to not utilize this approach. Why?

Response: We thank the reviewer for raising this question. While adding 2-perfluorohexylethyl acrylate and pentafluorophenyl acrylate to co-polymerize with PFPE-DMA can increase the dielectric constant of the film and potentially improves the performance of the fully stretchable transistor array, the overall uniformity of the films, after spincoated on the top of the dextran sacrificial layer, is not as good as that of the film made from 100% PFPE-DMA. Since the uniformity of the dielectric film strongly effects the yield of the transistor array, extensive optimization of the parameters (*e.g.*, solvent selection, spincoating rate, etc.) is required before applying this co-polymerization methods to make a large-area-uniform stretchable dielectric layer for the fully stretchable transistors array. Therefore, we decided to not apply this co-polymerization strategies to the fully stretchable transistor array in our first demonstration shown in current manuscript.

We have included an explanation to the revised manuscript as “*Notably, the uniformity of the PFPE-DMA/2-perfluorohexylethyl acrylate and PFPE-DMA/pentafluorophenyl acrylate thin films need to be further improved for fully stretchable TFT array fabrication.*”

Comment-3: (minor) Page 6: for the final device, the authors employ a bottom-gate configuration, but the initial evaluation of the dielectric is performed with a top-gate configuration. Why?

Response: We thank the reviewer for pointing this out. The reason to choose the top-gate configuration for the dielectric characterization is mainly to take advantage of the higher quality film that can be made from the spincoated semiconductor film on the modified silicon oxide substrate and the compatibility of spincoated perfluorinated dielectrics on top of the polymeric semiconductor layer. Since the quality of the semiconductor made through this method is more reproducible and it has been the standard method to test the performance of dielectric layers, we choose to use this high-quality semiconductor device to evaluate the dielectric properties of the dielectric material to avoid any variations from the polymeric semiconductor film. When fabricating the fully stretchable transistor array, we used inkjet printing to pattern semiconductor polymers over a large area. As a result, we needed to perform additional surface treatment of the PFPE-DMA film before inkjet printing. Therefore, bottom-gate configuration is chosen for the ease of fabrication process for the fully stretchable transistor array.

We have added an explanation in the revised manuscript to clarify our choice as “*To control the quality of spincoated polymer semiconductor for characterizing the dielectric performance of PFPE-DMA, we fabricated a top-gate-bottom-contact (TGBC) OFET to evaluate PFPE-DMA as a suitable stretchable dielectrics (Fig. 2a).*”

Comment-4: Paragraph spanning page 6-7: how can the mobility increase with decreasing dielectric thickness? To my understanding, the drain current will increase but not the mobility. The authors should explain how they anticipate that an increased vertical electric field will increase the mobility measured in the horizontal direction?

Response: We thank the reviewer for raising this question. This increase is mainly due to electric-field dependent mobility for semiconductor with traps. Given the polymer semiconductor film with many grain boundaries and various chain conformations that can trap charges, the carrier mobility in the semiconductor polymers is strongly dependent on the carrier density, which depends on the applied electric field (*see papers by* Tanase, C., Meijer, E. J., Blom, P. W. M. & de Leeuw, D. M. Unification of the hole transport in polymeric field-effect transistors and light-emitting diodes. *Phys. Rev. Lett.* **91**, 216601 (2013); Cho, J. H. *et al.* Printable ion-gel gate dielectrics for low-voltage polymer thin-film transistors on plastic. *Nature Mater.* **7**, 900-906 (2008)). Large carrier densities result in increased trap-filling and a general smoothing of electrostatic potential variations in the polymer film due to trapped charges, and these combined effects may lead to higher carrier mobilities. By reducing the thickness of the dielectrics, the vertical electric field will increase and thus increase induced charge carrier density and as a result increase the mobility measured in the horizontal direction.

We have added these citations in our manuscript as REF #34 and 35, and revised the manuscript as following: “According to previous reports^{34,35}, the carrier mobilities of semiconducting polymers are strongly dependent on the carrier density; a higher carrier density generally results in increased trap-filling and a more smoothing of electrostatic potential variations in the polymer film. These combined effects may lead to higher carrier mobilities. By reducing the thickness of the dielectrics, the vertical electric field will increase and thus increase the mobility measured in the horizontal direction. Upon reducing the thickness of the dielectric layer while measuring at the same applied gate voltage, the mobility of our thin film transistor increases as expected since the mobility of organic semiconductor thin film is typically electric-field dependent (Fig. 2c) as opposite to the behaviour of TFTs made of ionic dielectrics which do not show much thickness dependence^{30,36,37}.”

Comment-5: On the same topic, I do not understand how the claimed gate-dielectric thickness-dependence on the electronic mobility in the OSC can prove that ion motion in the dielectric is “minimal”. My intuition tells me the opposite, and I recommend that the authors instead measure, e.g., the time dependence of the off current (which likely will change if ions can traverse the dielectric-OSC interface).

Response: Thanks for the comments. Our previous papers (Kong, D. *et al.* Capacitance characterization of elastomeric dielectrics for applications in intrinsically stretchable thin film transistors. *Adv. Funct. Mater.* **26**, 4680–4686 (2016); Wang, C. *et al.* Significance of the double-layer capacitor effect in polar rubbery dielectrics and exceptionally stable low-voltage high transconductance organic transistors. *Sci. Rep.* **5**, 17849 (2015)) investigated effects of mobile ions on dielectric properties of stretchable gate dielectric layers. We found that double-layer capacitive effect dominated when the dielectric had relatively high dielectric constant (>4) and with some ionic impurities. Since double-layer capacitance is independent of the thickness of the gate dielectrics, the mobility of those transistors stays the same when significantly different thicknesses of gate dielectric layers were used. Therefore, the lack of gate-dielectric thickness-dependence on the electronic mobility in the semiconductor polymer measured from our current system indirectly suggests that ion motion in the dielectric is “minimal”. However, the more direct characterization of whether there is ionic effect is by frequency dependence capacitance measurement, which has been shown and discussed in **Figure 1g**.

We have cited these papers in our revised manuscript as REF #30 and 36. Thank you very much for your comments!

Comment-6: Page 7: it is stated that in order to pattern the OSC it is important that it features good ambient stability. Why?

Response: Thanks for raising this point. Since we carry out the inkjet printing process for semiconductor polymer patterning in ambient atmosphere, a good ambient stability is necessary.

We have added addition explanation in the revised manuscript as “.....possess good ambient stability as the printing is typically conducted at ambient environment.” Thank you very much for your comments!

Comment-7: On a number of occasions, the authors refer to the incorrect Figure, which must be corrected.

Response: We have changed all the incorrected references to figures in our revised manuscript. Thank you!

Comment-8: Fig. 2g: an explanation for the different measurement conditions should be included. For instance, what is the air interface?

Response: Thank you. We believe what you mean is the measurement conditions in **Fig. 2h**. We have included additional explanation of the different measurement conditions in revised **Fig. 2h** and **Fig. 2h**

caption as following: “*Top surface: top surface of PFPE-DMA film spincoated on dextran/SiO₂ substrate. Bottom surface: bottom surface of PFPE-DMA film spincoated on dextran/SiO₂ substrate. Bottom surface with modification: bottom surface of PFPE-DMA spincoated on PDMS-azide/dextran/SiO₂ substrate post-baked at 150 °C for 40.*” Thank you for your close read and suggestions!

Comment-9: Figs. 2i-j: the authors should clarify what the dashed lines indicate. On my screen and printout, the central grey region looks identical within and outside the dashed boundaries.

Response: we added arrows to clarify what the dashed lines indicate in our revised **Fig. 2i-j**. Thank you for your comments!

Comment-10: The LEC section is inadequate at this stage and must be rewritten. A few specific issues are described below.

Response: We thank the reviewer for raising this critique to our manuscript. We have added more detailed information and discussions into the LEC section as detailed below for **Comment-11**. We believe these revisions have substantially improved the manuscript. Thank you again for your close read and suggestions!

Comment-11: The LEC is distinguished by the existence of mobile ions in the active material, and the selection and concentration of this critical element should be described in the main text (although I note that it is mentioned in the experimental section). Similarly, the main text suggest that it is only one ion conductor in the active material, while the experimental section reveals two. The authors must correct.

Response: We have added following discussions and modifications. “*Figure 4a shows the device structure of a single-pixel OLEC. To prepare a stretchable OLEC, we used polyurethane acrylate coated by Ag nanowires (PUA-AgNWs) as the electrodes and PEDOT:PSS as the hole injection layer to sandwich the stretchable light-emitting layer (Fig. 4a, top). Here the PEDOT:PSS layer can effectively decrease the leakage current, enhancing the device performance. The light emission layer contains a blend of a light-emitting polymer (Super Yellow, SY), ion conducting polymer, ethoxylated trimethylpropane triacrylate (ETT-15) and lithium trifluoromethane sulphonate (LiTf) with the weight ratio of 20:20:2:1. Here we used the ion conducting polymer⁴¹ synthesized by our group instead of the commonly used polyethylene oxide (PEO) mainly because of its higher stretchability than PEO. ETT-15 serves as the ionically conductive component that can be polymerized by heating, and LiTf can provide the ionic dopant to form the PIN junction in the emission layer. The resulting OLECs can be stretched up to 30% strain without delamination or crack formation (Fig. 4b). To balance the stretchability while maintaining sufficient light emitting cell current density, the optimized weight ratio of SY to ionic conducting polymer was found to be about 1:1. The turn-on current density of the LEC is around 2 mA/cm² (Fig. 4c). The luminance ratio variation of the OLEC is increased with the applied voltage (Supplementary Fig. 7). The recording time is around 2 min. Importantly, the current density from a single OLEC device was stable when subjected to less than 30% strain (Fig. 4c, inset). The working mechanism of the OLEC is described below. When a voltage is applied to an OLEC, ions in the emission layer are redistributed to form an electrical-double layer at anode (PEDOT:PSS/PUA/AgNW) and cathode (PUA/AgNW) interfaces to allow easy hole or electron injection, respectively, leading to electrochemical doping to form a light-emitting PIN junction. The Supplementary Table S1 provides a summary of some related research works on PLEC with excellent flexibility and stretchability.*” Thank you for your suggestions!

Comment-12: The turn-on time of LEC devices can vary from a few tenths of a second to several hours depending on the active material composition (and the drive voltage). This is obviously a very critical

measure for the merit of an LEC in a display application, and the authors must include data on the temporal evolution of the current density and the luminance.

Response: The reviewer raised a good point about the relationship between current density and luminance. We totally agree with it. We have included the previously published results as **Response Figure 1** showing that the temporal dynamics of switching the LEC device is less than one second to a couple of milliseconds. Thanks a lot for your suggestions and understanding.

Response Figure 1. Time response of elastic OLECs containing similar composition. Transient light emission response under voltage pulse from elastic OLEC (Liang, J. *et al.*, “Elastomeric polymer light-emitting devices and displays” *Nature Photon.* **7**, 817 (2013)) comprised of yellow light-emitting polymer (SuperYellow), ethoxylated trimethylolpropanetriacrylate, polyethylene oxide and lithium trifluoromethane sulphonate (*left panel*), and elastic OLEC (Zhang, Z. *et al.*, “A colour-tunable, weavable fibre-shaped polymer light-emitting electrochemical cell” *Nature Photon.* **9**, 233 (2015)) comprised of blue light-emitting polymer (PF-B), ethoxylated trimethylpropane triacrylate and lithium trifluoromethane sulphonate (*right panel*). Notably, PF-B has built-in oligo (ethylene oxide) side groups that are beneficial for ionic conductivity and high electroluminescent performance, which is similar to our blended OLEC.

Comment-13: On a similar topic, as LECs are dynamic devices that change resistance during the in situ doping process, it is not meaningful to present data without disclosing under which conditions they were recorded (e.g. the time of recording should be included in Fig. 4c.).

Response: We thank the reviewer for the suggestion. We included the recording time in the revised manuscript as following: “*The whole recording time is around 2 min.*”

Comment-14: It is also notable that quantitative information on the measured efficiency and luminance is lacking, and the authors should include this missing piece of information. It would also be very interesting to learn more about how their LECs perform in comparison to the state-of-the-art for stretchable and non-stretchable LECs. After all, essentially identical but non-stretchable LEC devices void of the new ion transporter have been frequently reported.

Response: Since we don’t have access to a luminance meter to measure the intensity, we used a power meter to measure the emitted light intensity as a function of applied voltage. However, light loss due to waveguiding effect of the front substrate and reflection are not considered. We have included this data in a new **Supplementary Figure 8**. In addition, we have included a new **Supplementary Table S1**,

comparing some of performances of our LEC device to other reported stretchable and non-stretchable LECs.

Comment-15: Since the merit of a display obviously is (in part) defined by its peak light-emission intensity and pixel switching time, it would be interesting to be educated on the current status of the stretchable display as regards to these metrics. As I anticipate that parasitic capacitances and poor LEC performance could limit these metrics at this early stage of development, I emphasize that I am not expecting any record value, but more detailed information on the current performance would nevertheless be interesting.

Response: We have included light emitting intensity as a function of applied voltage in a new **Supplementary Figure 8**. In addition, we have included a new **Supplementary Table S1**, comparing some of performances of our LEC device to other reported stretchable and non-stretchable LECs. Thanks a lot for your suggestions and understanding.

Reviewer #2

Comment-1: The authors claim that fully stretchable AM displays are important to skin electronics for wearables. The presented work demonstrates, to the authors knowledge, for the first time a fully stretchable AMOLEC display. Although the work seems interesting, to my opinion, stretchable PMOLEC pixels have been demonstrated before in literature. In addition, OLED and AMOLED displays have shown to be fully flexible before, e.g. <https://doi.org/10.1002/jsid.547> by Samsung. Therefore, to my opinion, the novelty of the work is not the first AM display in the world.

Moreover, the authors demonstrate the technology with only a 3x2 matrix.

Response: We thank the reviewer point out the fully flexible AMOLED display paper (J.-H. Hong, *et al.* 9.1-inch stretchable AMOLED display based on LTPS technology. *J. Soc. Inf. Disp.* **25**, 194-199 (2017)). However, we want to respectfully point out there are fundamental differences between our work with this flexible AMOLED display work:

- None of the electronic components (e.g., OLED, TFTs, interconnects, substrates, etc.) of the fully flexible AMOLED display is made from stretchable materials. Importantly, their TFTs are made from polycrystalline silicon, a rigid and brittle electronic material. On the contrary, we report a series of material innovations, enabling the experimental demonstration of a fully stretchable AMOLEC are made entirely from fully stretchable materials.
- The reported fully flexible AMOLED display can only tolerate less than 5% strain. Notably, this level of the stretchability most likely comes from the plastic deformation of the PI substrate used in the paper (**Fig. 3** in J.-H Hong *et al.* *J. Soc. Inf. Disp.* **25**, 194-199 (2017)). On the contrary, due to the integration of fully stretchable optical and electronic materials, our stretchable AMOLEC can easily tolerate up to 30% stretchability with no plastic deformation. The OTFT array can even tolerate up to 100% stretchability.
- While as the first demonstration, there are many potentials for the future improvement in our techniques to increase the density and number of pixels, we want to respectfully point out that our current manuscript only claims that we have demonstrated the first fully stretchable AMOLEC array, not a display.

In conclusion, we believe that comparing to the mentioned paper, our claim of first AMOLEC array still sustain.

To further clarify this confusion point, we added this paper to our revised reference list and further included the discussion in the introduction section as following: “Some reports²⁴ that leveraged the small strain tolerance up to 5% of flexible but non-stretchable electronic and optical materials (e.g., polycrystalline silicon and commercially available OLED) to fabricate AMOLED. Their stretchability is however, impossible to be further improved due to the limited stretchability of these flexible electronic and optical materials”

We want to thank the reviewer’s comment again. We believe this change substantially improves the clarity and quality of our current manuscript.

Comment-2: Regarding Fig 3g, can the authors explain why the mobility drops and increases again with increased strain?

Response: We believe this may be due to the strain-induced alignment of semiconductor network together with perpendicular geometric change due to the Poisson effect. This effect sustained or even slightly increased the overall mobility of the transistor as shown from our previous reports (Xu, J. *et al.* Highly stretchable polymer semiconductor films through the nanoconfinement effect. *Science*, **549**, 59-64 (2017); Wang, G.-J. N. *et al.* Inducing elasticity through oligo-siloxane crosslinks for intrinsically stretchable semiconducting polymers. *Adv. Funct. Mater.* **26**, 7254-7262, (2016)).

Comment-3: The TFTs seems to operate at high voltages, however the OLEC at 10V. Which are the voltages during operation? To my opinion, this would also limit the 'skin' operation.

Response: We thank the reviewer point out this question. The fact that TFTs operate at relatively high voltages is due to the thick dielectric layer with low dielectric constant and the modest mobility of the semiconductor polymer (PII2T) that we used in this work. However, our group recently have developed a series of semiconductor polymers and dielectric layer which require much lower operation voltage e.g., <30 V (Wang, S. *et al.* Skin electronics from scalable fabrication of an intrinsically stretchable transistor array. *Nature* **555**, 83-88 (2018)). Further incorporation of these materials into our work will further facilitate the proposed “skin” operation.

We have added the related work in our revised reference list as REF #15.

Comment-4: The parameters of the OLED/TFT change after 10 cycles, I was wondering how many cycles are required for typical skin-electronics.

Response: We thank reviewer to raise this question. For skin-electronics applications, it is likely thousands to tens of thousands of cycles will be needed. Indeed, the cycle stability of our devices will need further improvement. In addition to improving intrinsic material stability over strain cycles, interfacial adhesion between such multi-layer device is extremely important, which will require substantial interfacial engineering efforts in future work.

Reviewer #3

General Comment: The authors presented an intrinsically stretchable AM-OLEC array fabricated by solution-based inkjet printing. This is an extension work of the printed stretchable OTFT reported by the same group. I think the major improvement from the previous work is the studies on the dielectric layer and the fabrication of the active matrix LEC. Within my knowledge, this work is the first demonstration

of the printed stretchable AM-OLEC display, which is a valuable advance in the printed electronics society. I recommend publication of this work after revising a few comments.

Response: We thank the reviewer for the positive comments on our paper!

Comment-1: Detailed description on the transistor fabrication looks somewhat decreasing the importance of the stretchable dielectric layer for printing. I suggest, emphasize the dielectric part and the other transistor fabrication can be reduced.

Response: We thank the reviewer for the close read and suggestions. We have revised the manuscript through expanding the discussion of dielectrics and condensing the discussion of transistor fabrications. We thank the review for this thoughtful suggestion which greatly improved the quality of the manuscript.

Comment-2: In the dielectric layer, what is the minimum thickness in the aspect of the film stability and in the aspect of device operation (leakage). More details about the dielectric layer (such as capacitance, break-down voltage, thickness dependence, etc..) will be helpful to the readers.

We thank the reviewer for the suggestion. We have measured the break-down voltage and their thickness dependence, which is included as new **Supplementary Figure 2**. Specifically, the new measurement shows that the PFPE-DMA film can sustain >100 V breakdown voltage when the thickness is even less than 200 nm.

Reviewers' Comments:

Reviewer #1:

Remarks to the Author:

The authors have responded appropriately to my input (although I would have preferred more detail as regards to the LEC performance), and my recommendation is thus that the manuscript can be accepted.

Reviewer #2:

Remarks to the Author:

The authors have submitted a thoroughly revised manuscript for publication in Nature Communications. The quality of the manuscript has been improved. My queries from previous review have been answered.

I do have some additional questions, not hampering publication of this manuscript, but more out of curiosity:

- in Fig. 3, the authors show the mobility and on-current versus strain and cycles. I was wondering, as another key parameter, how the leakage current of the device is changing under those conditions? This is an important parameter for the long-term and correct behavior of the device.
- There is a difference in parameters between parallel and perpendicular placed transistors with respect to the stretch direction. Is this hampering a fully 2D stretch for the application and limits the scope of the matrix to 1D? There is no discussion on this in the current text.

Reviewer #3:

Remarks to the Author:

The authors addressed well the comments raised by this reviewer. I recommend publication of this work.

Point-by-point response letter to reviewers' comments

Reviewer #2

Comment-1: In Fig. 3, the authors show the mobility and on-current versus strain and cycles. I was wondering, as another key parameter, how the leakage current of the device is changing under those conditions? This is an important parameter for the long-term and correct behavior of the device.

Response: We thank the reviewer for raising this question. We have added the data related to leakage current as new **Supplementary Figure 9**. Specifically, the new data shows that when stretched to 100% strain or cycled to 1,000 times, the thin film transistor array still has stable leakage current compared with the initial condition, which demonstrates that the device possesses stable long-term performance.

Comment-2: There is a difference in parameters between parallel and perpendicular placed transistors with respect to the stretch direction. Is this hampering a fully 2D stretch for the application and limits the scope of the matrix to 1D? There is no discussion on this in the current text.

Response: We thank the reviewer for raising this question. The integrated matrix can also be stretched in both directions up to 30% strain, however, we only included the figures with stretching in one direction. Due to the impact from current pandemic, we are unfortunately unable to retake the pictures for the stretched matrix. We appreciate the understanding from the reviewers and editor.